# Evaluation of Pesticide Residues in Vegetables from the Asir Region, Saudi Arabia

**DOI:** 10.3390/molecules25010205

**Published:** 2020-01-03

**Authors:** Mohamed F. A. Ramadan, Mohamed M. A. Abdel-Hamid, Montasser M. F. Altorgoman, Hamed A. AlGaramah, Mohammed A. Alawi, Ali A. Shati, Hoda A. Shweeta, Nasser S. Awwad

**Affiliations:** 1Pesticide Analysis Research Department, Central Agriculture Pesticide Laboratory, Agriculture Research Centre, Dokki 12618, Giza, Egypt; drfatihi81@gmail.com; 2Abha Food Safety Laboratory, Asir Municipality, Abha 61421, Saudi Arabia; MohamedMedhat@alexu.edu.eg (M.M.A.A.-H.); MontasserAltorgoman@alexu.edu.eg (M.M.F.A.); 3Department of Chemistry, Faculty of Science, Alexandria University, Alexandria 21524, Egypt; 4Department of Biochemistry, Faculty of Science, Alexandria University, Alexandria 51521, Egypt; 5Research Centre for Advanced Materials Science (RCAMS), King Khalid University, P.O. Box 9004, Abha 61413, Saudi Arabia; hamedsa@hotmail.com; 6Inspection and License Department Asir Municipality, Abha 61421, Saudi Arabia; mohdawi@hotmail.com; 7Biology Department, Faculty of Science, King Khalid University, Abha 9004, Saudi Arabia; aaalshati@kku.edu.sa; 8College of Pharmacy, Umm Al-Qura University, P.O. Box 715, Makkah 21955, Saudi Arabia; hashweeta@uqu.edu.sa

**Keywords:** pesticide residue, MRL, vegetables, Asir, UHPLC-MS/MS, monitoring

## Abstract

This study’s aim was to determine the pesticide residues in 10 different vegetable commodities from the Asir region, Saudi Arabia. We evaluated 211 vegetable samples, collected from supermarkets between March 2018 and September 2018, for a total of 80 different pesticides using ultrahigh-performance liquid chromatography–tandem mass spectrometry (UHPLC-MS/MS) and gas chromatography–tandem mass spectrometry (GC-MS/MS) after extraction with a multi-residue method (the QuEChERS method). The results were assessed according to the maximum residue limit (MRL) provided by European regulations for each pesticide in each commodity. All lettuce, cauliflower, and carrot samples were found to be free from pesticide residues. A total of 145 samples (68.7%) contained detectable pesticide residues at or lower than MRLs, and 44 samples (20.9%) contained detectable pesticide residues above MRLs. MRL values were exceeded most often in chili pepper (14 samples) and cucumber (10 samples). Methomyl, imidacloprid, metalaxyl, and cyproconazole were the most frequently detected pesticides. Based on the results of this study, we recommend that a government-supported program for the monitoring of pesticide residues in vegetables be established to promote consumers’ health and achieve sustainable farming systems.

## 1. Introduction

Maintaining high agricultural output requires the use of pesticides, since, in high-input agricultural production systems, pests, among other crop invaders, including herbs and fungi, inevitably need to be managed [1]. However, reliance on pesticides is unsustainable due to their harmful effects on the environment and human health. The risk to human health comes from direct or indirect exposure to pesticide residues in primary or derived agricultural products [2]. Pesticides play a role in many human health problems, and can exert acute effects, such as dizziness, headaches, rashes, and nausea, and chronic effects, such as cancers, neurotoxicity, genotoxicity, birth defects, impaired fertility, and endocrine system disruption [3]. Children are particularly susceptible to exposure to pesticides [4]. Consequently, governments of different countries have enacted legislation in order to reduce consumer exposure to harmful pesticides, and regulate the appropriate use of pesticides in terms of the authorization that is granted, the type of registration (application rates and pre-harvest intervals), and allowing for free deliberation as to which products are to be treated with pesticides as long as the treatment complies with the established maximum residue limits (MRLs) [5]. For a specific pesticide applied to a certain food item, there is a tolerance level that, when exceeded, is called ‘violative residue’. Commonly, violation takes place when residues that exceed the established tolerance for a specific food item are detected. Tolerances may be not an accurate standard for health-related levels, but are at least suitable for the maximum residue limits that have been set for the use of pesticides by law [6]. Furthermore, violation rates do not consider the degree of consumption of various food items and the existing levels of pesticide residues [7].

The detection of pesticide residues in vegetable commodities, for the purpose of optimally evaluating vegetables’ quality and mitigating potential risks to human health, is a predominant aim of pesticide research. The most common extraction procedure for a wide range of pesticide classes is the Quick, Easy, Cheap, Effective, Rugged, and Safe (QuEChERS) method. In this method, liquid–liquid extraction (LLE) with salting-out (MgSO_4_ and NaCl salts) is first performed, followed by a cleanup using primary secondary amine (PSA)-bonded silica with dispersive solid phase extraction (dSPE). This method was proposed for the extraction of pesticide residues from food commodities [8]. Gas chromatographic and Liquid chromatographic methods coupled with mass spectrometric detection (GC-MS/MS and LC-MS/MS, respectively) are among the most highly selective and sensitive instruments for determining the residues of pesticides in a variety of food commodities. They also allow for a simultaneous quantitative and qualitative analysis of the targeted analytes and have excellent separation efficiency and a high speed of analysis. Several multi-residue methods, and selective and sensitive detectors, for detecting different classes of pesticides with different chemical and physical properties and separating individual compounds have been proposed [9,10,11]. There is a limited amount of information about the contamination of food, particularly vegetables, with pesticide residues in the Asir region, Saudi Arabia. There is no published literature on the contamination of vegetables with pesticide residues in Asir, which is of concern when taking into consideration the fact that vegetables are prone to being contaminated with higher pesticide levels when compared to other food groups [12]. Thus, the purpose of this study was to monitor pesticide residues in vegetables collected from supermarkets in the Asir region in order to establish a database that includes the levels of these residues in this region. We employed highly sensitive and selective multi-residue methods for the quantitative and qualitative determination of pesticides from several compound classes with different chemicals and physical properties using GC-MS/MS and LC-MS/MS. Then, we evaluated whether the results complied with existing regulations, particularly the European ones. Finally, we considered the appropriateness of the studied commodities for human consumption with respect to the official MRLs.

## 2. Results

### 2.1. Verification of the Analytical Method

The procedure for extracting multi-residue pesticides in vegetable samples was carried out using the rapid, sensitive, and rugged QuEChERS method. The method was validated under optimal conditions by investigating the recovery, precision, and detection limits. The recovery values at two fortification levels ranged from 70.5% to 126.6%, and the precision values (expressed as RSD, %) were below 20% for all of the investigated analytes (Table 1), which satisfies the criteria for quantitative methods for pesticide residues in food [13]. The limit of detection (LOD) and limit of quantitation (LOQ) were calculated by multiplying the standard deviation of repeatability by factors of 3 and 6, respectively [14]. All pesticide LOD (0.0004–0.0023 mg kg^−1^) and LOQ (0.0008–0.0047 mg kg^−1^) (Table 1) values were less than the maximum residue levels (MRLs) appointed for each analyte in each commodity. In this study, 80 pesticides from different chemical classes were deemed to be among those that are commonly used in vegetable production in Saudi Arabia. A total of 51 pesticides were analyzed by LC-MS/MS, and the remainder were analyzed by GC-MS/MS.

### 2.2. Evaluation by Commodity

The concentrations of pesticide residues in 211 vegetable samples from the Asir region, southwest Saudi Arabia, were determined. Detectable residues were found in 145 samples (68.7%), while 66 samples (31.3%) were found to be residue-free. The percentage of detected residues was high for all analyzed vegetables except carrot, cauliflower, and lettuce. All samples of cucumber (100%) and chili pepper (100%) were contaminated with pesticide residues, while none of the carrot, cauliflower, and lettuce samples contained pesticide residues. Only 3.9% of tomato samples, 10% of cabbage samples, 15% of eggplant samples, 18.2% of potato samples, and 25% of onion samples were pesticide-free. Cucumber (100%), chili pepper (100%), tomato (96.1%), and cabbage (90%) had the highest percentage of detected residues (Table 2).

### 2.3. The Frequency of Detection and Exceedance of MRLs

Pesticide residue concentrations above the MRLs stipulated by EU regulations [15] were detected in a total of 44 samples (20.9%). MRL values were surpassed most often in chili pepper and cucumber; 50% of the chili pepper samples and 41.7% of the cucumber samples were found to contain pesticide residue concentrations above the MRL values. Table 3 presents the frequency and ranges of the detectable residues in the tested commodities.

### 2.4. Evaluation by Pesticide Residue

In this study, the concentrations of 80 different pesticides were determined in 10 different vegetable commodities. Of the 80 pesticides, 37 were detected in the tested samples. Of the detected substances, 20 were insecticides (54.1%), 12 were fungicides (32.4%), 4 were herbicides (10.8%), and 1 was a growth regulator (2.7%). Thirty percent (30%) of the detected insecticides (6 of 20) exceeded the MRL, and the insecticide methomyl was found to most frequently exceed the MRL. Of the detected fungicides, 41.7% (5 of 12) exceeded the MRL, and the fungicide cyproconazole was found to most frequently exceed the MRL. Of all detected pesticides, methomyl, imidacloprid, metalaxyl, cyproconazole, carbendazim, triadimenol, profenofos, chlorpyrifos-methyl, malathion, and acetamiprid were found the most often. Figure 1 shows the detection frequency of the pesticides that frequently occurred in the analyzed samples.

As shown in Figure 1, methomyl was the most frequently detected pesticide in all tested commodities. Residues of methomyl were detected in tomato, chili pepper, cucumber, cabbage, onion, potato, and eggplant in the concentration range 0.005–0.307 mg kg^−1^ and exceeded the MRL in all of these commodities except for tomato and potato, which contained residues at or below the MRL values. Imidacloprid was the second most frequently detected pesticide in the vegetable commodities and was found in the concentration range 0.014–0.199 mg kg^−1^. Residues of imidacloprid were found in tomato, cucumber, cabbage, onion, eggplant, and potato; however, they did not exceed the MRLs in any of these commodities. Metalaxyl was detected in tomato, potato, cucumber, and chili pepper in the concentration range 0.007–0.419 mg kg^−1^, and exceeded the MRL values in only tomato and potato. Residues of cyproconazole and carbendazim were detected in cucumber, chili pepper, eggplant, and cabbage in the concentration range 0.008–0.541 mg kg^−1^ and 0.004–0.158 mg kg^−1^, respectively. Cyproconazole exceeded the MRLs in all four of these commodities, while carbendazim exceeded the MRLs only in cabbage (a concentration of 0.158 mg kg^−1^). Triadimenol and chlorpyrifos-methyl were found in cabbage, onion, potato, eggplant, chili pepper, and tomato in the concentration range 0.004–0.044 mg kg^−1^ and 0.004–0.061 mg kg^−1^, respectively. Profenofos exceeded the MRLs in cabbage and chili pepper with a concentration of 0.496 mg kg^−1^ and 0.041 mg kg^−1^, respectively. Profenofos was also detected in tomato; however, the concentration was within the MRL. Malathion and myclobutanil were detected in eggplant, cabbage, and cucumber in the concentration range 0.007–0.273 mg kg^−1^ and 0.010–0.470 mg kg^−1^, respectively. Myclobutanil exceeded the MRL in eggplant with a concentration of 0.470 mg kg^−1^ and in cucumber with a concentration of 0.436 mg kg^−1^. Malathion exceeded the MRL only in cucumber with a concentration of 0.273 mg kg^−1^. Chlorantraniliprole and tebuconazole exceeded the MRLs in potato with a concentration of 0.031 mg kg^−1^ and 0.039 mg kg^−1^, respectively. Chlorfenapyr exceeded the MRLs in cucumber and chili pepper with a concentration of 0.034 mg kg^−1^ and 0.026 mg kg^−1^, respectively. Ethion was detected only in chili pepper and exceeded the MRL with a concentration of 0.061 mg kg^−1^. Acetamiprid residues were found to fall within the MRL in tomato, chili pepper, and eggplant. Diazinon residues were found to fall within the MRL in chili pepper and cucumber. Additionally, measurable residues of hexaconazole were detected in tomato, chili pepper, cucumber, and potato. Of all detected pesticides, the highest concentration levels were found in chili pepper (0.541 mg kg^−1^, cyproconazole), cabbage (0.496 mg kg^−1^, profenofos), cucumber (0.436 mg kg^−1^, myclobutanil), tomato (0.419 mg kg^−1^, metalaxyl), and eggplant (0.307 mg kg^−1^, methomyl).

### 2.5. The Co-Occurrence of Pesticide Residues

The incidence of multiple residues in the tested commodities is shown in Figure 2. Of the tested commodities, 12.8% (27 samples) contained a single residue, 41.7% (88 samples) contained two residues, 10.4% (22 samples) contained three residues, and 3.79% (eight samples) contained four residues. The presence of multiple pesticide residues was observed most frequently in chili pepper, tomato, cucumber, potato, cabbage, and eggplant (Figure 3).

## 3. Discussion

This study, to our knowledge, is the first to monitor the concentration of 80 pesticide residues in different vegetable commodities from the southwest region of Saudi Arabia. Saudi Arabia’s southwest region is considered to be an important agricultural area due to its fertile ground, suitable climate, and torrential rain throughout the year. The three main agricultural areas in Saudi Arabia’s southwest region are located in Jizan, Baha, and Asir [16]. In this study, we tested 211 vegetable samples for pesticide residues. Of all tested samples, 66 samples (31.3%) were found to be residue-free, while 145 samples (68.7%) were found to contain a detectable amount of pesticide residue. Of the analyzed samples, 20.9% contained pesticide residues whose concentration exceeded the MRLs. Similarly, Osman et al. (2010) analyzed 160 vegetable samples collected from supermarkets in the Al-Qassim region, Saudi Arabia and found that 44.4% of the tested samples were free of pesticide residues, 55.6% contained detectable amounts of pesticide residues, and 59.6% (53 of 89) of the pesticide-contaminated samples had a residue concentration greater than the MRL values. Also, Jallow et al. (2017) analyzed 150 vegetable and fruit samples from Kuwait and found that 42% of the tested samples were residue-free, 58% contained a detectable amount of residue, and 21% contained pesticide residues whose concentration was greater than the MRL values. The incidence of pesticide residues in the tested vegetables may be due to vegetable crops being damaged by many pests and their various species [17,18] (Table 4); therefore, different pesticides are applied to protect these crops against pests and diseases, particularly vegetable crops that are cultivated under greenhouse conditions [19,20]. The humid conditions and large amount of food in greenhouse environments make them ideal habitats for pests and make crops in these environments more susceptible to pests such that successive applications of pesticide treatments are required to prevent considerable crop losses [21,22].

The highest concentrations of detected pesticides were recorded for the fungicide cyproconazole (in chili pepper), followed by the insecticide profenofos (in cabbage), the fungicide myclobutanil (in cucumber), the fungicide metalaxyl (in tomato), and the insecticide methomyl (in eggplant). The pesticide residue levels were found to vary among the vegetable types, and are greatly dependent on the harvest time, size of the fruit, and pesticide application mechanism [23,24,25]. Cyproconazole most frequently exceeded the MRL values (10 samples), followed by methomyl (nine samples), metalaxyl (eight samples), profenofos (five samples), chlorfenapyr (three samples), myclobutanil and ethion (two samples), and malathion and chlorantraniliprole (one sample). MRLs are typically set by using a scientific risk assessment [26] and dominate pesticide residue standards, which may differ from one country to another [27] due to different agricultural and climatic conditions and directly reflect the pesticide application rate [28]. MRL exceedance may be due to GAP non-compliance, cross-contamination or spray drift, contamination from a previous use of persistent pesticides, and/or unexpectedly slow degradation of residues [29]. Cyproconazole is a broad-spectrum fungicide and acts as a sterol biosynthesis inhibitor (a demethylation inhibitor) in fungi. It has moderate mobility in soil (K_Foc_ = 173–711 mL g^−1^), moderate to high persistence in soil (DT_50_ = 72.4–347 days), and high residue stability. Cyproconazole has moderate acute toxicity when inhaled and is very highly toxic to organic organisms. The FAO/WHO set the ADI to 0.02 mg/kg bw/day and the ARfD to 0.06 mg/kg bw with a safety factor (SF) of 100 [30,31]. Methomyl is an oxime carbamate and works by inhibiting acetylcholinesterase (AChE) enzymes. The overuse of methomyl may be due to its effectiveness as a contact and systemic broad-spectrum insecticide against organophosphorus-resistant pests and foliar treatment. It also has very high mobility in soil (K_Foc_ = 13.3–42.8 mL/g), low to moderate persistence in soil (DT_50 lab 20 °C_ = 4.6–11.5 days), high solubility in water, and high stability. However, it was classified by the EPA as a restricted-use pesticide (RUP) due to its high acute toxicity to humans. The European Food Safety Authority (EFSA) and FAO/WHO set the ADI, ARfD, and NOAEL of methomyl to 0.0025 mg/kg bw with a safety factor (SF) of 100 [32,33,34]. In the present study, the MRL values were exceeded most often in chili pepper (14 samples), cucumber (10 samples), tomato (five samples), potato (five samples), cabbage (four samples), and eggplant (four samples). All of the tested commodities were cultivated in Saudi Arabia except for chili pepper, which was imported mainly from India. Among the tested samples, chili pepper was found to be the most highly contaminated commodity that exceeded the MRL. On May 2014, the ministry of agriculture in Saudi Arabia decided to ban the import of chili pepper from India after detecting a high level of pesticide residue in this commodity. Saudi Arabia lifted the ban after confirmation that exporters had complied with regulations on the permissible levels of pesticide residues in chili pepper. High levels of contamination with pesticide residues may be due to overuse of pesticides to control pests and/or farmers having a lack of awareness about pesticide application doses, mechanisms, and standard pre-harvest intervals (PHIs). Additionally, the non-availability of proper guidance about pesticides’ application, inadequate supervision by relevant departments, and non-compliance with best agricultural practices may lead to contaminated vegetables, which are considered to be a potential source of health hazards to consumers [35,36]. Household processing is needed to reduce the intake of pesticide residues. Washing, the most prevalent form of processing, can more effectively remove water-soluble pesticides than low-polarity materials. Peeling can also be used to reduce pesticide residue intake, particularly the intake of non-systemic pesticides that remain in the peel [37,38].

In terms of pesticide residues, some vegetables were found to contain more than one type of residue, particularly those vegetables that were cultivated under greenhouse conditions, which require consecutive applications of pesticides. In recent years, the decrease in pests’ susceptibility to pesticides has led to changes in the global chemical pesticide market and widespread use of mixtures, such as binary pesticide mixtures. Insufficient knowledge about the proper use of pesticides, a lack of awareness about integrated pest management (IPM) methods, and a desire to increase the attractiveness of a product may be additional reasons for the harmful co-occurrence of pesticide residues [39]. The occurrence of multiple residues does not entail non-compliance with MRL legislation if the individual pesticide concentrations do not exceed permissible limits. The existing law does not establish limits for those cases where pesticides co-occur. However, products with multiple pesticide residues should be evaluated carefully in order to be sure that a combination of pesticides was not used intentionally to circumvent MRL limits on single substances. The EFSA developed a software tool, called the Monte Carlo risk assessment (MCRA) tool, that is able to assess the cumulative risks arising from exposure to multiple pesticides [40]. From a toxicological viewpoint, if it has not been observed that the incidence of multiple residues could have additive or synergic effects, they may still affect the overall quality of the food. The quality index for residue (IqR) can be used to evaluate how multiple residues affect the quality of the commodity [41,42,43]. The IqR is calculated as the sum of the ratios between the residue concentrations and the corresponding MRLs (Equation (1)):(1)$$IqR=\overset{n}{\underset{i=1}{\sum }}\left(Concentration_{i}/MRL_{i}\right).$$

This index considers the ratio of residue concentrations to the allowable limits in order to observe the degree of contamination as compared to the MRLs (see Figure 4). The Iqr divides the quality of fruit and vegetables into four groups: optimal (IqR = 0), good (IqR 0–0.6), adequate (IqR = 0.6–1), and inadequate (IqR > 1). The results presented in Table 5 show that 31.28%, 22.27%, and 15.17% of the tested samples were of optimal, good, and adequate quality, respectively, while 31.28% of the tested samples were of inadequate quality.

The excessive use of pesticides in Saudi agriculture, particularly in greenhouse crop production, is a serious problem. Precedence should be given to improving strategies for the reduction of pesticides in agriculture through tighter government regulations, including the implementation of laws in relation to pesticide use, the control of pesticide sales, adherence to pesticide label instructions, the application of appropriate pre-harvest intervals, compliance with integrated pest management approaches, and best agricultural practices [44,45]. Organic farming may be an effective and safe way to reduce excessive pesticide use. In April 2005, Saudi Arabia started an organic farming project in cooperation with the Research Institute of Organic Agriculture (FiBL) and the German Society for International Cooperation (GIZ). The project’s aim was to develop a functioning and sustainable organic farming sector. According to the GIZ report, the southwest region is a reduced organic surface region [46]. Therefore, the Saudi organic farming association (SOFA) should implement programs that help farmers convert to organic farming, which is a holistic and environmentally friendly agricultural production system.

## 4. Materials and Methods

### 4.1. Chemicals and Reagents

Pesticide active ingredients were obtained from Dr. Ehrenstorfer GmbH (Augsburg, Germany) with certified purities greater than 95%. The monitored pesticides, their classification [47,48], and technical data for the LC-MS/MS pesticides and the GC-MS/MS pesticides are listed in Table 6 and Table 7, respectively. As shown in Figure 5, the set of selected pesticides includes most insecticides. As the standards have different purities, the concentration was corrected individually for each one. Methanol and acetonitrile (pesticide-grade) were obtained from Fischer company, Dallas, TX, USA. Ultra-pure deionized water (18 MΩ cm) was obtained from a water purification system (PURELAB Option-R, ELGA, BUCKS, UK). Magnesium sulfate (MgSO_4_), sodium chloride (NaCl), Sodium Citrate, disodium citrate sesquihydrate, PSA, and graphite carbon black (GCB) were obtained from Agilent (Santa Clara, CA, USA).

### 4.2. Preparation of Intermediate, Working Solutions, and Calibration Curves

By dissolving a corrected weight of each compound (according to its purity) into 10 mL of acetonitrile, standard stock solutions were prepared at 1000 mg kg^−1^. An intermediate mix of standards with a concentration of 5 mg L^−1^ was then prepared. Lastly, the working standard solutions were used to prepare matrix-matched calibrations between 2.5 and 200 μg L^−1^.

### 4.3. Sample Collection

According to the 2002/63/EC [49] regulation, a total of 211 different vegetable samples covering 10 commodities that are frequently consumed by local people (tomato, cucumber, cabbage, eggplant, chili pepper, onion, potato, carrot, lettuce, and cauliflower) were collected from supermarkets in Asir, Saudi Arabia in the period from March 2018 to September 2018. These samples were transported under cold conditions to the laboratory and kept at 4 °C. Shortly after their arrival, they were analyzed for pesticide residues following the QuEChERS method described below.

### 4.4. LC-MS/MS Analysis

LC-MS/MS analysis was conducted using a liquid chromatograph (Thermo ultimate 3000, Dionex Softron GmbH, Rohrbach, Germany) combined with a triple quadruple mass detector with a heated electrospray ionization (HESI) source (Thermo, TSQ Quantum Access Max, San Jose, CA, USA) and a Thermo Scientific Hypersil GOLD aQ column (100 × 2.1 mm; 1.9 μm particles). Time-specific SRM (t-SRM) windows were used at the target compound’s retention time to maximize the performance of the mass spectrometer. The sheath gas flow rate was 55 units, the AUX gas flow rate was 15 units, the capillary temperature and the heater temperature were 280 °C and 295 °C, respectively, the spray voltage was 3500 V, and the cycle time was 0.2 s. Water containing 0.1% formic acid and 4 mM ammonium formate (mobile phase A) and methanol containing 0.1% formic acid and 4 mM ammonium formate (mobile phase B) were used for the gradient program, which started with 2% B and sharply increased to 30% B over 0.25 min, then linearly increased to 100% B over 19.75 min, and finally maintained 100% B for 6 min. The column was then reconditioned to 2% B for 4 min. The column’s temperature was set at 40 °C. The injection volume was 10 μL at a flow rate of 0.3 mL/min. At least two multi-reaction monitoring (MRM) transitions were monitored for each compound.

### 4.5. GC-MS/MS Analysis

All samples were analyzed using a TSQ Quantum XLS GC-MS/MS system equipped with a Thermo Scientific TRACE GC Ultra gas chromatograph with a programmable split/splitless injector. The capillary column was a Thermo Scientific TRACE TR-Pesticide II (30 m × 0.25 mm × 0.25 µm) with a 5 m guard column. Sample volumes of 1.0 μL were injected in split/splitless injection mode, and a deactivated fused-silica liner with a diameter of 2 mm was used. The temperature of the injection port was set at 240 °C (isothermal). A constant velocity of 1 mL/min was used for the helium carrier gas. The oven temperature program was initially set to hold at 80 °C for 1 min, then ramp with no hold to 140 °C at 25 °C/min, and finally ramp to 200 °C with no hold at 5 °C/min. The oven program’s total length was 39 min with an injection-to-injection time of 10 min. The transfer line and the ion source of the mass spectrometer were heated to 280 °C. A higher-level standard was used to optimize transitions in the positive electron ionization (EI)-SRM mode on the TSQ Quantum XLS GC-MS/MS. The t-SRM function tool allows one to monitor SRM transitions more effectively by monitoring only the analyzed compounds at specific elution times, allowing for partial overlap. The collision gas (Argon) pressure was 1.2 mTorr, and the Q1/Q3 resolution was 0.7 u (full width at half maximum (FWHM)). Electron ionization was set at −70 eV and the emission current was 30 µA.

### 4.6. Extraction Procedure

The acetate-buffered QuEChERS method was applied to determine the concentration of pesticides in the vegetable samples (AOAC 133 Official Method 2007.01) [50]. Homogenization for more than 1 min was carried out using a blender (Waring, DCA, Torrington, CT, USA) to obtain thoroughly mixed homogenates. A 15 g portion of the homogenized sample was weighed in a 50 mL PTFE tube and 15 mL of acetonitrile containing 1% acetic acid was added. Then, 6 g of MgSO_4_ and 2.5 g of sodium acetate trihydrate were added and the sample was shaken for 4 min. The sample was then centrifuged at 4000 rpm for 5 min (Eppendorf 5804 R, Hamburg, Germany) and 5 mL of the supernatant was transferred to a 15 mL PTFE tube containing 750 mg MgSO_4_ and 250 mg PSA. Furthermore, graphitized carbon was used to clean up the chili pepper (10). The extract was shaken for 20 s using a vortex mixer and then centrifuged for 5 min at 4000 rpm. Approximately 3 mL of the supernatant was filtered through a 0.45 μm PTFE filter (13 mm in diameter).

### 4.7. Quality Control

Recovery tests were done using blank samples that were free from pesticide. Subsamples of those blanks from the different studied commodities were spiked with two levels (0.010 and 0.1 mg kg^−1^) of each compound. Then, they were extracted in accordance with the above-described QuEChERS procedure. Recovery and precision (expressed as RSD, %) were measured by analyzing three samples of each commodity individually.

## 5. Conclusion

This study presented evidence of the incidence of pesticide residues in vegetable commodities from the southwest region of Saudi Arabia. The most highly contaminated commodities were found to be chili pepper and cucumber. Methomyl, imidacloprid, metalaxyl, and cyproconazole were the most frequently detected pesticide residues in the tested commodities. The high observed levels of pesticide residues may represent a potential health risk for consumers. As most of these vegetables are consumed raw, household processing, including washing, peeling, and cooking, is necessary in order to reduce the amount of pesticide residues in them. Based on our findings, we recommend that pesticide residues in a greater number of crops be regularly monitored over long periods in order to better protect consumers’ health.

## Figures and Tables

**Figure 1 molecules-25-00205-f001:**
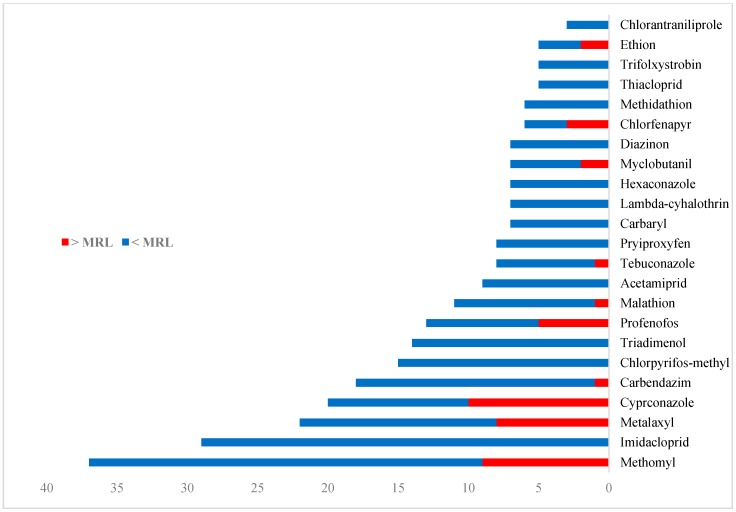
Frequency of the most-often-detected pesticides in the analyzed samples.

**Figure 2 molecules-25-00205-f002:**
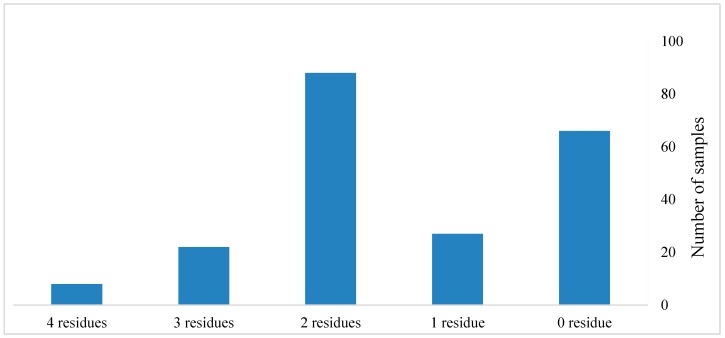
The co-occurrence of pesticide residues in the tested samples.

**Figure 3 molecules-25-00205-f003:**
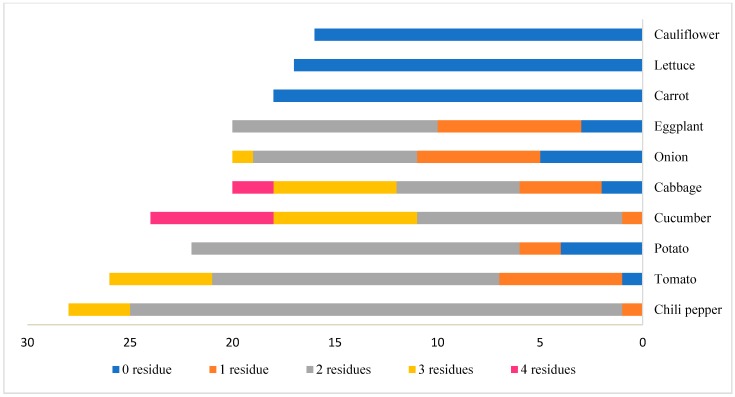
The occurrence of multiple residues in different vegetables.

**Figure 4 molecules-25-00205-f004:**
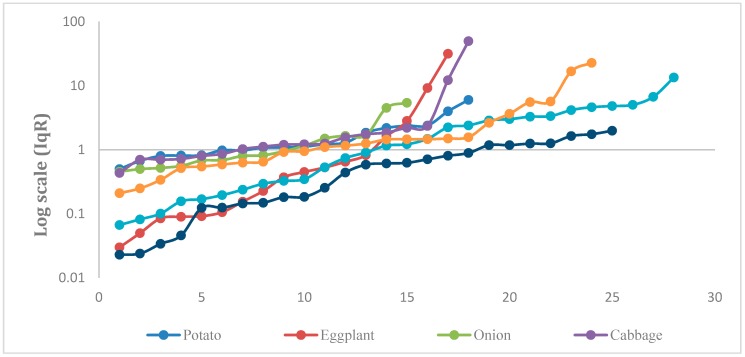
The calculated quality index for residue (IqR) for the selected vegetable commodities on a Log scale.

**Figure 5 molecules-25-00205-f005:**
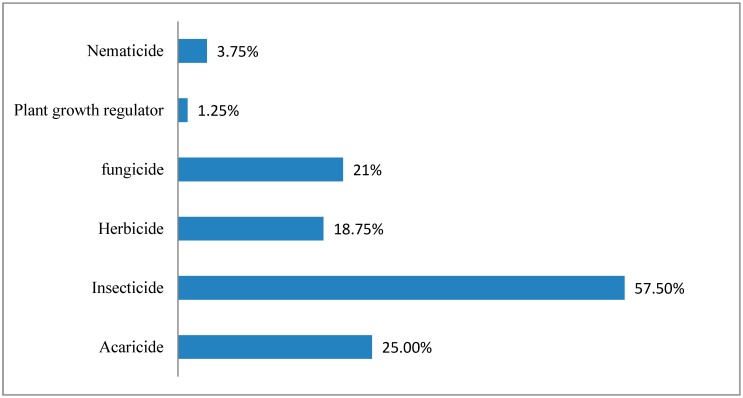
Distribution of selected pesticides according to usage.

**Table 1 molecules-25-00205-t001:** Recovery, precision, and detection limit ranges for selected pesticides that exceeded the maximum residue levels (MRLs) in different commodities.

Pesticide Name	Recovery Range at 0.01 mg kg^−1^	RSD % Range at 0.01 mg kg^−1^	Recovery Range at 0.1 mg kg^−1^	RSD % Range at 0.1 mg kg^−1^	LOD Range mg kg^−1^	LOQ Range mg kg^−1^
Carbendazim	83–103	2–8	92–112	1–10	0.0006–0.0021	0.0012–0.0042
Chlorantraniliprole	74–98	3–8	89–100	3–10	0.0008–0.0023	0.0016–0.0045
Chlorfenapyr	76–115	2–8	82–108	2–11	0.0006–0.0023	0.0012–0.0047
Cyproconazole	79–118	2–7	84–123	0.5–14	0.0005–0.002	0.0009–0.004
Ethion	76–103	3–8	80–104	1–11	0.0009–0.0021	0.0018–0.0042
Malathion	75–120	3–7	78–123	1–16	0.0008–0.002	0.0016–0.004
Metalaxyl	78–117	2–7	75–121	2–16	0.0005–0.002	0.001–0.0039
Methomyl	75–91	5–7	82–90	1–11	0.0011–0.0018	0.0022–0.0037
Myclobutanil	82–117	2–9	89–118	4–14	0.0006–0.0023	0.0012–0.0045
Profenofos	76–118	3–7	80–108	1–16	0.0008–0.002	0.0016–0.004
Tebuconazole	76–117	3–8	95–122	2–13	0.0009–0.0023	0.0018–0.0045
min–max range	74–120	2–9	75–123	0.5–16	0.005–0.0023	0.0009–0.0047

**Table 2 molecules-25-00205-t002:** Frequency of samples with pesticide residues in the Asir region, Saudi Arabia from March 2018 to September 2018.

Commodity	No. of Analyzed Samples	Residue-Free Samples	Samples with Residue > LOD	Samples with Residue < MRL	Samples with Residue > MRL
Cucumber	24	0 (0%)	24 (100%)	14 (58.3%)	10 (41.7%)
Chilli pepper	28	0 (0%)	28 (100%)	14 (50%)	14 (50%)
Tomato	26	1 (3.9%)	25 (96.1%)	20 (76.9%)	5 (19.2%)
Cabbage	20	2 (10%)	18 (90%)	14 (70%)	4 (20%)
Eggplant	20	3 (15%)	17 (85%)	13 (65%)	4 (20%)
Potato	22	4 (18.2%)	18 (81.8%)	13 (59.1%)	5 (22.7%)
Onion	20	5 (25%)	15 (75%)	13 (65%)	2 (10%)
Carrot	18	18 (100%)	0 (0%)	0 (0%)	0 (0%)
Lettuce	17	17 (100%)	0 (0%)	0 (0%)	0 (0%)
Cauliflower	16	16 (100%)	0 (0%)	0 (0%)	0 (0%)
Total number	211				
Residue-free		66 (31.3%)			
Total > LOD			145 (68.7%)		
Total < MRL				101 (47.9%)	
Total > MRL					44 (20.9%)

**Table 3 molecules-25-00205-t003:** Pesticide concentration ranges, frequencies, and MRLs in the analyzed vegetable samples.

Commodity	No. of Samples with Residues < MRL (%)	No. of Samples with Detectable Residues > MRL (%)	Detected Pesticide	Frequency	No. of Samples > MRL	Range Min–Max mg Kg^−1^	MRL (mg Kg^−1^)
Tomato	20 (76.9%)	5 (19.2%)	Buprofezin	2		0.023–0.124	1
			Chlorantraniliprole	2		0.017–0.031	0.6
			Hexaconazole	3		0.003–0.005	0.01
			Imidacloprid	10		0.043–0.116	0.5
			Acetamiprid	4		0.012–0.137	0.5
			Metalaxyl-M	8	5	0.023–0.419	0.3
			Methidathion	3		0.006–0.015	0.02
			Methomyl	7		0.005–0.008	0.01
			Profenofos	3		0.095–0.231	10
			Pyriproxyfen	4		0.033–0.167	1
			Triadimenol	2		0.017–0.044	0.3
			Chlorpyrifos-methyl	1		0.061	1
			Lambda-Cyhalothrin	1		0.017	0.07
Cucumber	14 (58.3%)	10 (41.7%)	Carbendazim	6		0.013–0.083	0.1
			Clethodim	2		0.04–0.113	0.5
			Cyproconazole	7	4	0.023–0.123	0.05
			Diazinon	3		0.004–0.009	0.01
			Difenoconazole	4		0.019–0.097	0.3
			Hexaconazole	1		0.004	0.01
			Imidacloprid	7		0.071–0.199	1
			Metalaxyl-M	4		0.013–0.083	0.5
			Methomyl	5	2	0.009–0.222	0.01
			Metribuzin	1		0.039	0.1
			Myclobutanil	3	1	0.028–0.436	0.2
			Penconazole	2		0.015–0.026	0.1
			Tebuconazole	4		0.091–0.159	0.6
			Triadimenol	1		0.009	0.15
			Trifloxystrobin	3		0.016–0.063	0.3
			Malathion	6	1	0.011–0.273	0.02
			Chlorfenapyr	3	2	0.007–0.034	0.01
			Cypermethrin	1		0.017	0.2
			Chlorbufam	1		0.005	0.01
			Cyfluthrin	1		0.014	0.1
			Kresoxim-methyl	2		0.009–0.017	0.05
			Lambda-Cyhalothrin	1		0.023	0.05
Chili pepper	14 (50%)	14 (50%)	Acetamiprid	3		0.031–0.054	0.3
			Clethodim	2		0.019–0.043	0.5
			Cyproconazole	5	4	0.008–0.541	0.05
			Diazinon	3		0.013–0.026	0.05
			Ethion	5	2	0.007–0.061	0.01
			Hexaconazole	2		0.005–0.008	0.01
			Hexythiazox	1		0.029	0.5
			Metalaxyl-M	4		0.033–0.0103	0.5
			Methomyl	7	3	0.005–0.199	0.04
			Metribuzin	1		0.011	0.1
			Penconazole	1		0.027	0.2
			Profenofos	7	4	0.007–0.041	0.01
			Pyriproxyfen	2		0.043–0.056	1
			Carbendazim	4		0.022–0.098	0.1
			Tebuconazole	1		0.017	0.6
			Thiacloprid	1		0.009	1
			Triadimenol	1		0.027	0.5
			Trifloxystrobin	2		0.014–0.035	0.4
			Chlorfenapyr	3	1	0.004–0.026	0.01
			Chlorpyrifos-methyl	2		0.007–0.051	1
			Cypermethrin	1		0.111	0.5
			Kresoxim-methyl	2		0.015–0.021	0.8
Cabbage	14 (70%)	4 (20%)	Carbaryl	3		0.005–0.006	0.01
			Carbendazim	3	1	0.006–0.158	0.1
			Cyproconazole	2	1	0.043–0.255	0.05
			Diazinon	1		0.009	0.01
			Fenarimol	1		0.011	0.02
			Forchlorfenuron	2		0.005–0.007	0.01
			Hexythiazox	1		0.039	2
			Imazapyr	1		0.008	0.01
			Imidacloprid	3		0.014–0.051	0.5
			Kresoxim-methyl	1		0.023	0.1
			Methidathion	3		0.008–0.013	0.02
			Methomyl	5	1	0.006–0.071	0.01
			Myclobutanil	2		0.010–0.017	0.05
			Penconazole	1		0.015	0.05
			Profenofos	3	1	0.007–0.496	0.01
			Triadimenol	3		0.004–0.007	0.01
			Malathion	3		0.007–0.013	0.02
			Chlorpyrifos-methyl	3		0.005–0.008	0.01
			Lambda-Cyhalothrin	2		0.027–0.031	0.15
Onion	13 (65%)	2 (10%)	Buprofezin	1		0.018	0.05
			Dimethoate	2		0.005–0.007	0.01
			Carbaryl	4		0.009–0.015	0.02
			Forchlorfenuron	1		0.005	0.01
			Methomyl	5	2	0.009–0.054	0.01
			Triadimenol	3		0.005–0.008	0.01
			Chlorpyrifos-methyl	3		0.004–0.008	0.01
			Lambda-Cyhalothrin	2		0.019–0.031	0.2
			Imidacloprid	4		0.019–0.053	0.1
Eggplant	13 (65%)	4 (20%)	Carbendazim	5		0.033–0.121	0.5
			Chlorpyrifos-methyl	3		0.017–0.026	1
			Cyproconazole	3	1	0.031–0.141	0.05
			Imidacloprid	1		0.045	0.5
			Kresoxim-methyl	1		0.017	0.6
			Malathion	2		0.009–0.013	0.02
			Myclobutanil	2	1	0.016–0.47	0.3
			Thiacloprid	4		0.013–0.051	0.7
			Triadimenol	1		0.039	0.3
			Acetamiprid	2		0.015–0.102	0.2
			Methomyl	3	2	0.008–0.307	0.01
			Lambda-Cyhalothrin	1		0.017	0.3
Potato	13 (59.1%)	5 (22.7%)	Chlorantraniliprole	2	1	0.015–0.031	0.02
			Metalaxyl-M	5	3	0.013–0.079	0.02
			Methomyl	5		0.005–0.010	0.01
			Tebuconazole	3	1	0.011–0.039	0.02
			Triadimenol	3		0.005–0.007	0.01
			Chlorpyrifos-methyl	3		0.004–0.008	0.01
			Imidacloprid	4		0.031–0.076	0.5
			Cyproconazole	3		0.015–0.022	0.05
			Hexaconazole	1		0.006	0.01
			Cyfluthrin	1		0.021	0.04
			Chlorbufam	1		0.009	0.01
			Pyriproxyfen	2		0.013–0.026	0.05

**Table 4 molecules-25-00205-t004:** Common vegetable crop pests.

**Host**	**Aphids**	**Armyworms and Cutworms**	**Maggots and Colorado Potato Beetles**	**Thrips**	**Loopers**	**Slug and Spider Mites**
Chili pepper	*Myzus persicae*	*Spodoptera exigua*, *Mamestra configurata*	-	-	*Autographa californica*	*Tetranychus* spp. (mite)
Cucumber	*Myzus persicae*	*Agotis ipsilon*, *Peridroma saucia*	*Delia platura* (maggot)	*Frankliniella occidentalis*, *Frankliniella williamsi*	*Autographa californica*, *Trichoplusia ni*	*Tetranychus* spp. (mite)
Tomato	*Myzus persicae*, *Macrosiphum euphorbiae*	*Spodoptera exigua*, *Mamestra configurata*	*Leptinotarsa decemlineata* (beetle)	-	*Macrosiphum euphorbiae*	*Tetranychus* spp. (mite)
Cabbage	*Brevicoryne brassicae*	*Spodoptera exigua*, *Mamestra configurata*	*Delia brassicae*	*Frankliniella occidentalis*, *Frankliniella williamsi*	*Autographa californica*	*Milax gagates* (slug)
Eggplant	*Myzus persicae*	-	*Leptinotarsa decemlineata* (beetle)	-	-	-
Potato	*Macrosyphum euphorbiae*, *Myzus persicae*	*Mamestra configurata Walker*, *Xestra c-nigrum Linnaeus*	*Leptinotarsa decemlineata* (beetle)	*Thrips tabaci*, *Frankliniella occidentalis*	*Autographa californica*, *Trichoplusia ni Hubner*	*Deroceras reticulatumr* (slug), *Tetranychus* spp. (mite)
Onion	-	*Spodoptera exigua*, *Mamestra configurata*	*Delia antiqua*, *Delia platura* (maggot)	*Thrips tabaci*, *Frankliniella occidentalis*	-	-
Carrot	*Myzus persicae*	*Agotis ipsilon*, *Peridroma saucia*	-	-	-	-
Lettuce	*Nasonovia ribisnigri*, *Pemphigus bursarius*	*Spodoptera exigua*, *Mamestra configurata*	-	-	*Autographa californica*	*Milax gagates* (slug)
Cauliflower	*Myzus persicae*	*Spodoptera exigua*, *Mamestra configurata*	*Delia brassicae* (maggot)	*Frankliniella occidentalis*, *Frankliniella williamsi*	*Autographa californica*	*Milax gagates* (slug)
**Host**	**Wireworms**	**Whitefly and Diamondback Moths**	**Garden Symphylans**	**Cucumber Beetles and Imported Cabbageworms**	**Flea Beetles and Carrot Flies**	
Chili pepper	*Limonius* spp.	*Trialeurodes vapariorum* (whitefly)	*Scutigerella immaculata*	-	*Epitrix subcrinita* (beetle)	
Cucumber	*Limonius* spp.	-	*Scutigerella immaculata*	*Acalymma trivittatum* (beetle)	-	
Tomato	*Limonius* spp.	*Trialeurodes vapariorum* (whitefly)	-	-	*Epitrix tuberis Gentner* (beetle)	
Cabbage	*Ctenicera* spp., *Limonius* spp.	*Plutella xylostella* (moth)	*Scutigerella immaculata*	*Pieris rapae* (worm)	*Phyllotreta cruciferae* (beetle)	
Eggplant	*Limonius* spp.	*Trialeurodes vapariorum* (whitefly)	-	*Tetranychus* spp. (beetle)	*Epitrix subcrinita* (beetle)	
Potato	*Ctenicera* spp., *Limonius* spp.	*Trialeurodes vapariorum* (whitefly)	*Scutigerella immaculata L*	*Diabrotica undecimpunctata Linnaeus* (beetle)	*Epitrix tuberis Gentner* (beetle)	
Onion	*Limonius* spp.	-	-	-	-	
Carrot	-	-	*Scutigerella immaculata*	-	*Psila rosae* (carrot fly)	
Lettuce	*Limonius* spp.	-	-	*Acalymma trivittatum* (beetle)	-	
Cauliflower	*Ctenicera* spp., *Limonius* spp.	*Plutella xylostella* (moth)	*Scutigerella immaculata*	*Pieris rapae* (worm)	*Phyllotreta cruciferae* (beetle)	

**Table 5 molecules-25-00205-t005:** The quality of the selected vegetables according to the calculated IqR.

	Optimal (IqR: 0)	Good (IqR: 0–0.6)	Adequate (IqR: 0.6–1)	Inadequate (IqR: > 1)
Cucumber		6	4	14
Chili pepper		11	2	15
Tomato	1	13	5	7
Cabbage	2	1	6	11
Eggplant	3	11	2	4
Potato	4	1	8	9
Onion	5	4	5	6
Carrot	18			
Lettuce	17			
Cauliflower	16			
Total	66	47	32	66
Percentage, %	31.28	22.27	15.17	31.28

**Table 6 molecules-25-00205-t006:** Summary of LC-MS/MS pesticides (properties and use).

SN	Pesticide Name	Group	Use ^a^	R_t_	Precursor Ion	Transition 1 (Quantity)	CE	Transition 2	CE	Transition 3	CE
1	Acetamiprid	Neonicotinoid	I	3.87	223.1	126.1	20	90.2	36		
2	Atrazine	Triazine	H	9.32	216	174	17				
3	Bifenazate	Carbazate	A, I	16.02	301.23	170	19	152	37		
4	Buprofezin	Thiadiazin (chitin synthesis inhibitor)	A, I	20.78	306.21	201	12	116	18		
5	Cadusafos	Organophosphorous	I, N	20.21	270.97	158.9	16	97	36		
6	Carbaryl	Carbamate	A, PR, I	8.13	202.08	145	10	127	31		
7	Carbendazim	Benzimidazole carbamate	F	2.75	192.1	160.06	18	132.1	30		
8	Clethodim	Cyclohexene oxime (cyclohexane dione)	H	21.38	360.19	164	22	268	12		
9	Chlorantraniliprole	Anthrailic diamide	I	11.79	482.13	450.89	19	283.81	17		
10	Chlorpyrifos	Organophosphorus	I	23.81	350	198	16	97	33		
11	Cyproconazole	Triazole	F	15.58	292.13	125	30				
12	Desmetryn	Methylthiotriazine	H	6.63	214.11	172.07	16	82.21	28	57.34	30
13	Diazinon	Organophosphorous	A, I, N	18.51	305.03	169.1	23	153.13	21		
14	Diethofencarb	Carbanilate	F	12.43	268.21	226	13	180.1	18		
15	Difenoconazole	Triazole	F	21.12	406.17	251	23	111	52		
16	Dimethoate	Organophosphorus	A, I	3.68	230.11	199.1	10	125.1	22		
17	Emamectin	Avermectin	I	24.99	886.7	158	30	302	18		
18	Ethion	Organophosphorus	A, I	23.56	384.92	142.97	27	97.09	46		
19	Famoxadone	Oxazole	F	20.08	392.11	331.22	7	238.03	186		
20	Fenamiphos	Organophosphorus	N	17.47	304.03	217.01	22	234.03	6		
21	Fenarimol	Pyrimidine	F	16.32	331.12	268	22	81	35		
22	Forchlorfenuron	Phenylurea (Growth stimulator)	PG	10.77	248.14	129	17	93	26		
23	Hexaconazole	Conazole(triazole)	F	19.39	314.14	70.2	21	159	18		
24	Hexythiazox	Thiazolidine Carboxamide	A	24.03	353.24	228.2	16	168.1	24		
25	Imazapyr	Imidazolinone	H	9.64	262.06	216.98	18	201.97	25		
26	Imidacloprid	Neonicotinoid	I	3.29	256.12	209.1	19	175.1	20		
27	Indoxacarb	Oxadiazine	I	21.9	528.3	203	36	293	14		
28	Isoproturon	Phenylurea	H	10.09	207.1	72	18	165.15	14		
29	Kresoxim-methyl	Strobilurin	F	17.77	314.07	267.14	9	222.13	15		
30	Linuron	Phenylurea	H	12.87	249.1	182	19	160	18		
31	Metalaxyl	Amide(anilide)	F	10.36	280.11	220.1	18	192.1	15		
32	Methidathion	Thiadiazole organothiophosphate	I, A	10.92	302.9	85.2	22	144.92	4		
33	Methomyl	Oxime carbamate	A, I	2.63	163.05	106.1	8	88.1	11		
34	Metribuzin	Triazinone	H	6.23	215.09	187.07	16	130.97	15		
35	Myclobutanil	Triazole	F	15.58	289.13	125	30	70.2	20		
36	Penconazole	Triazole	F	18.43	284.12	159	32	70.1	16		
37	Pendimethalin	Dinitroaniline	H	24.1	282.09	212	10	194.11	15	119.07	23
38	Primicarb	Carbamate	I	4.59	239.09	182	18	72	22		
39	Profenfose	Organophosphorous	A, I	22.05	372.9	302.8	20	143.86	33	127.97	40
40	Propiconazole	Triazole	F	18.91	342.2	159	30	69.2	22		
41	Pymetrozin	Pyridine	I	2.18	218	105	24	79	28		
42	Pyriproxyfen	Hormone Mimic	I	23.49	322.22	96	15	185.3	25		
43	Sethoxydim	Cyclohexene oxime (cyclohexane dione)	H	7.58	328	178	18				
44	Spinosyn A	Spinosyn	I	21.19	732.5	142	36	98	44		
45	Spinosyn D	Spinosyn	I	22.6	746.5	142	33	98	44		
46	Spiromesifen	Tetronic acid	A, I	24.73	371.3	273.3	14	255.3	25		
47	Tebuconazole	Triazole	F	18.57	308.22	70.2	20	125	32		
48	Tepraloxydim	Cyclohexene oxime (cyclohexane dione)	H	8.38	340	220	32	248	15		
49	Thiacloprid	Neonicotinoid	I	4.68	253.13	126.1	20	90.2	35		
50	Triadimenol	Triazole	F	14.26	296.1	70	15				
51	Trifloxystrobin	Strobilurin	F	21.54	409.3	186	20	206.1	15		

**^a^** I: Insecticide, A: Acaricide, F: Fungicide, H: Herbicide, PG: Plant Growth regulator, N: Nematicide.

**Table 7 molecules-25-00205-t007:** Summary of GC-MS/MS pesticides (properties and use).

SN	Pesticide name	Group	Use	R_t_	Parent	F1	CE	Parent	F1	CE
1	Bifenthrin	Pyrethroid	I, A	22.3	180.77	164.92	20	181.05	166.05	15
2	Bromophos ethyl	Organothiophosphate	I	18.46	358.41	284.48	30	358.41	302.57	17
3	Bromophos methyl	Organothiophosphate	I	17.24	328.9	313.8	14	331	315.76	13
4	Carbophenothion	Organothiophosphate	I, A	21	120.8	64.83	7	199	142.9	10
5	Fenchlorfos (Ronnel)	Organothiophosphate	I	15.3	284.91	269.92	13	286.91	271.91	20
6	Chlorbufam	Carbanilate	H	12.58	152.73	89.88	17	152.73	124.82	14
7	Chlorfenapyr	Pyrrole	I, A	19.84	246.71	226.7	13	246.711	199.45	25
8	Chlorpyrifos-ethyl	Organophosphorus	I, A	16.59	196.96	168.96	15	198.96	170.96	15
9	Chlorpyrifos-methyl	Organophosphorus	I, A	14.9	285.52	240.56	20	285.52	270.57	17
10	Cyanophos	Organophosphorus	I	12.97	242.69	108.83	10	242.69	126.84	7
11	Cyfluthrin	Pyrethroid	I	25.09	162.68	90.92	13	165.02	91.01	15
12	Cyhalothrin	Pyrethroid	I	23.37	180.8	151.71	25	197.04	141.03	13
13	Cypermethrin-1	Pyrethroid	I	25.34	162.67	90.86	13	180.78	151.53	20
14	Cafenstrole	Triazole	H	25.57	100.04	72.03	13	188.08	119.05	15
15	Deltamethrin	Pyrethroid	I	28	181	151.73	17	253	171.58	7
16	Diflufenican	Anilide	H	21.63	265.71	217.88	17	265.71	237.77	12
17	Esfenvalerate	Pyrethroid	I	27.13	124.85	88.97	16	167.05	125.04	10
18	Etofenprox	Pyrethroid ether	I	25.81	162.87	106.87	17	162.87	134.84	8
19	Fenamidone	Imidazole	F	19.03	224.01	125.01	15	224.01	196.01	10
20	Fenitrothion	Organophosphorus	I	16.02	124.76	78.94	7	276.66	259.84	7
21	Fenpropathrin	Pyrethroid	A, I	22.49	97.1	55.1	6	181	151.9	22
22	Fenthion	Organothiophosphate	I	16.96	277.64	108.85	17	278	169	14
23	Fenvalerate	Pyrethroid	A, I	26.75	124.82	88.94	20	167.05	125.04	10
24	Fluazifop-butyl	Aryloxyphenoxypropionate	H	20.05	282	91.2	18	282	238.1	16
25	Malathion	Organophosphorus	A, I	16.38	126.8	98.91	7	172.8	98.86	13
26	Procymidone	Dicarboximide	F	18.15	95.9	53	16	95.9	67.1	8
27	Propyzamide	Benzamide	H	13.16	172.69	108.81	25	172.69	144.7	13
28	Resmethrin	Pyrethroid	I	21.8	122.88	80.95	10	171.11	128.08	9
29	Sulfotep	Organothiophosphate	I, A	11.19	321.57	145.5	20	321.57	201.83	10

**^a^** I: Insecticide, A: Acaricide, F: Fungicide, H: Herbicide, N: Nematicide.

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
