# Peer review of "Evaluation of Pesticide Residues in Vegetables from the Asir Region, Saudi Arabia"

_molecules, 2020, doi:10.3390/molecules25010205_

Round 1
Reviewer 1 Report
This manuscript reported the assessment of pesticide residues in several vegetables from an area of Saudi Arabia. Basically it is a food analysis paper, with a focus on pesticide analysis, and it is definitely within the scope of Molecules. The analytical chemistry part is pretty straightforward, QuEChERS plus the LC/GC-MS analysis is a traditional way nowadays. Based on the results of the component and content analysis of the various pesticide residues, it is concluded that chili peppers and cucumbers are the two most affected vegetables, and the top pesticide overdosed are also listed. Although the samples were only from Asir region from Saudi Arabia, thus may not have a large impact to the whole country, at least it is a typical sample and may bring the attentions to people from that area, therefore, as far as I am concerned, this manuscript can be assigned as minor revision. The only concern I have is the sample collection. As it is stated in the text, the 211 samples were collected from big supermarkets between March and September in 2018. Is that the rain season of the area? If not, may the pesticide content be varied by the weather or season? or why analyze the samples collecting in those months, are the samples typical enough to represent the vegetables in the area? “Samples collecting from March to September” seems not that clear to me.
Author Response
Article : Manuscript ID: molecules-680182
Evaluation of Pesticide Residues in Vegetables from the Asir Region, Saudi Arabia
It has been reviewed by experts in the field and i request that make
minor revisions before it is processed further.
Please see the attachment.
I have only one comment related to the manuscript need extensive English revision and was done

Reviewer 2 Report
There are some editorial errors, mainly in relation to units, capital letters, etc.
I suggest combining Table 1 and 2 to make them more readable. Authors can add some information about toxicity of tested pesticides. Did Authors use the internal standards?
Line 134 The Authors stated, that: “…in split/splitless injection mode, incorporating a 2 mm i.d. deactivated fused-silica liner”. Liner was from silica? not glass?
Why did the Authors use linear velocity during GC analysis? This option of gas flow in the column caused an increasing reduction in flow at high temperatures and makes results impossible to reproduce.
How many repetitions were performed during the recovery study?
What about the toxicity of pesticides detected at higher levels than MRL’s values? The Authors discussed only the properties of these chemicals. These pesticides are typically detected substances in this region? They are correlated with the data regarding usage? What with the consumer risk assessment?
Author Response
Article : Manuscript ID: molecules-680182
Evaluation of Pesticide Residues in Vegetables from the Asir Region, Saudi Arabia
Please see the attachment
I have only one comment related to the manuscript need extensive English revision and was done
